# Psychological States and Training Habits during the COVID-19 Pandemic Lockdown in Spanish Basketball Athletes

**DOI:** 10.3390/ijerph18179025

**Published:** 2021-08-27

**Authors:** Jorge Lorenzo Calvo, Miriam Granado-Peinado, Alfonso de la Rubia, Diego Muriarte, Alberto Lorenzo, Daniel Mon-López

**Affiliations:** 1Facultad de Ciencias de la Actividad Física y del Deporte (INEF-Departamento de Deportes), Universidad Politécnica de Madrid, C/Martín Fierro 7, 28040 Madrid, Spain; jorge.lorenzo@upm.es (J.L.C.); alfonso.delarubia@upm.es (A.d.l.R.); diego.muriarte@upm.es (D.M.); alberto.lorenzo@upm.es (A.L.); 2Facultad de Educación y Psicología, Universidad Francisco de Vitoria (UFV), Ctra. Pozuelo-Majadahonda Km 1.800, 28223 Pozuelo de Alarcón, Spain

**Keywords:** basketball, psychological state, training habits, COVID-19

## Abstract

The outbreak of the COVID-19 pandemic and the resulting restrictions designed to slow the spread of infection greatly disrupted people’s lives. The present study aimed to investigate the impact of lockdown on the psychology, training, and sleep habits of a cohort of basketball players. An online survey involving 169 professional and amateur athletes was conducted using four validated psychological questionnaires (WLEIS-S, POMS, BRS, SMS-II) and a Likert scale to measure the rating of perceived exertion (RPE) and training variables. Gender differences in fatigue (*p* = 0.022); friendships (*p* = 0.017); others’ emotional appraisal (*p* < 0.001); and resilience (*p* = 0.031) were apparent, with higher values for women in all categories bar resilience. Comparisons before and during the lockdown revealed that all participants reduced their RPE (*p* < 0.001); training days (*p* = 0.004); and training hours (*p* < 0.001), and experienced a decline in the quality of sleep (*p* < 0.001). Sleep hours (*p* < 0.001) increased during lockdown. The professionals and females maintained their training days (*p* > 0.05), while the non-professionals and males did not. Psychological states during lockdown were a predictor of the differences in training and recovery variables. In situations where training and competition are limited, it is important to develop plans to maintain physical activity, good quality sleep, and promote greater emotional management and understanding to control negative moods.

## 1. Introduction

In 2020, a virus called Coronavirus 2 (SARS [severe acute respiratory syndrome]-CoV-2 or COVID-19) caused an unexpected pandemic [1]. Countries responded in a variety of ways to contain it. The Spanish government declared a state of alarm on 15 March, putting its cities in lockdown. This reduced people’s mobility and forced the closure of sports facilities and training centres [2].

Athletes were unable to continue training or competing as usual, regardless of their sporting level. The last official round (23) of the basketball league before lockdown was played on 7–8 March 2020, and from that point, the players’ lifestyles and training regimes became subject to several changes. The subsequent period of physical inactivity, which is typical of team sports during the off-season, would have negative effects on their fitness and mental well-being [3].

The pandemic affected training variables. The scientific literature shows that there was a significant decrease in the volume and intensity of practice in both professional and semi-professional sport [4]. Pillay et al. [5] analysed a sample of 692 athletes and discovered that most trained at a moderate intensity for a shorter time (30–60 min) with a lower working load. This led to a significant reduction in weekly energy expenditure [6].

In an attempt to maintain the level of physical performance after assessing the risk associated with a lack of activity or detraining, team staffs adapted training seasons [7,8]. Programmes or applications incorporating global positioning systems (GPS) to report the most relevant external load data for the physical activity and individual questionnaires were used to obtain information about external training loads and athletes’ rating of perceived exertion (RPE).

However, mandatory individualised training methodologies fail to reproduce the specific conditions associated with team sports, which require the involvement and interaction of teammates and opponents [9,10]. This has been evidenced in football and handball research [4,11,12]. A decline in associated prosocial behaviours means less enjoyment, effort, performance [13], and discrepancies in training intensity and volume. Moreover, the specific role of each player has to be considered in training days because in-game workloads vary [14,15] in terms of gender [16], performance level, intensity (e.g., moderate or high), and intermittence [17,18].

Physical inactivity caused by lockdown led to certain psychological disorders. Several studies have shown a deterioration in mental health amongst the populace [19,20,21,22]. Anxiety, depression, and post-traumatic stress disorder were some of the most reported variables during the pandemic [4]. Moreover, being a woman has been associated with adverse effects such as mental fatigue in several studies [23,24,25].

The lockdown had negative effects on emotional or mental states in some athletes [7]. However, it is not clear whether these conditions can be correlated with the type of sport (individual or collective), gender (male or female), or level (professional or non–professional). The earliest studies on athletes and COVID-19 [26] concluded that there were no differences between individual and collective sports, although Rubio et al. [27] showed higher levels of somatic symptoms in team players and faster rates of adaptation to isolation. Women tended to present higher scores for perceived stress than men [28,29] and worse functional psychosocial states. Di Fronso et al. [26] found no significant differences between professional and non-professional athletes.

However, it seems that, in professional sport, the detraining period did not necessarily lead to negative consequences. Non-specific training raised the athletes’ mood [6] and well-being [30], partly because it allowed them to recover from injury. It has been reported that athletes in training presented lower levels of depression and anxiety than non-athletes [31,32].

Rest and recovery times were affected by the pandemic [5]. Studies have shown that sleep is an important factor in basketball performance [33,34]. At least in the sporting context, this variable was affected by the pandemic [35]. Fox et al. [36] highlighted the possible relationship between a better subjective quality of sleep with positive individual match performance, mainly in offensive ratings and player efficiency. Mah et al. [37] concluded that more sleep and its associated habits can influence athletes’ performance, reaction time, and mood.

The present study analyses basketball players’ training and psychological variables such as motivation [38], anxiety levels and resilience [39], mood states [40], and emotional intelligence [41] before and after lockdown. These can all be related to performance [42] and self-determined motivation [43]. Most of the research conducted during the COVID-19 pandemic involved athletes from different sports, though a small number of studies [4,12] used participants from a specific sport to compare training and different psychological aspects. It is important to carry out more of these. Additionally, the unexpected and complex nature of the current situation has made it difficult to propose strategies adapted to the needs of athletes. Knowing how the confinement has affected their mental health and sporting performance could be valuable for similar situations in the future. The purpose of the present study, which was to evaluate physical and psychological variables amongst basketball players of both sexes and at different levels of performance, was undertaken in this context.

## 2. Method

### 2.1. Participants

Only basketball players in the Spanish Basketball Federation who were playing in one of the national leagues while the present study was being carried out were included. A total of 183 questionnaires were collected. Responses from injured players (*n* = 12), players infected with COVID-19 (*n* = 1) during the survey, and players who gave contradictory answers (*n* = 1) were rejected. The final sample comprised completed questionnaires from 169 players (aged 24.78 ± 6.38 years) who were confined (34.52 ± 3.7 days). Of the total number of participants, 122 were men (25.05 ± 6.78 years) and 47 women (24.09 ± 5.21 years), 34 professionals and 88 non-professionals, and 20 professionals and 27 non-professionals, respectively. Professional players were considered to be those players who participated in the first and second national leagues (ACB and LEB Gold for men and Liga Femenina 1 and 2 for women), while amateur players were those who played in the minor national leagues (LEB Silver, EBA and 1ª Nacional for men, and 1ª Nacional for women).

The gender distribution was considered optimal because the participants were a fit with the Spanish basketball player population. Men comprised 66.27% and women 33.73% of the sample; the Spanish basketball player population comprised 65.5% men and 34.5% women [44]. Additionally, the final number of participants could be considered a good data set in light of previous similar studies [4,12]. Finally, five players had previously been infected with COVID-19 and when they filled out the questionnaire they were already fully recovered from the disease. Descriptive variables of training variables and psychological variables are shown in Table 1. The study was approved by the ethics committee of the Polytechnic University of Madrid.

### 2.2. Instrument and Variables

The survey was divided into three sections following previous studies [4,12]: Demographic (Q1-10), training (Q11-20), and psychological (Q21-80). Demographic questions which were specific for basketball sport (sport level and play position) were adapted from Mon-López et al. [4,12] by two PhD professional basketball coaches. The demographic variables were age (years); gender (male or female); number of days confined (days); residence (Spain or other country); sport level (professional or non-professional); injured (yes or no); personal relationship with COVID-19 (no relation, COVID-19 infected, or COVID-19 recovered); play position; number of people with whom player was living during the lockdown); place of residence (i.e., where player was living during the lockdown by m^2^ (1 = 31 to 50; 2 = 51 to 70; 3 = 71 to 90; 4 = 91 to 110; 5 = more than 111). All demographic questions were single choice.

Training variables were volume (training days and training hours per week); intensity (RPE, 10-point Likert scale); and recovery (sleep hours and sleep quality, 10-point Likert scale). All the training variables referred to both before and during lockdown. Pre-isolation data were considered to be the usual training values for the players.

The psychological part of the survey comprised four areas: emotional intelligence (EI); mood states; resilience; and motivation. Emotional intelligence was measured using the Spanish validated version of the Wong Law Emotional Intelligence Scale Short form (WLEIS-S) [45]. This instrument consists of 16 items measuring four aspects of EI: self-emotion appraisal (SEA); others’ emotion appraisal (OEA); use of emotion (UOE); and regulation of emotion (ROE). Each item was measured on a Likert-type scale from 1 (totally disagree) to 7 (totally agree). The results were: SEA (α = 0.884); OEA (α = 0.816); UOE (α = 0.862); ROE (α = 0.889); and WLEIS-S (α = 0.871).

Moods were measured using the Spanish validated version of the Profile of Mood States (POMS) questionnaire [40] which consists of six mood subscales: tension; anger; depression; fatigue; vigour; and friendliness, with five items on each subscale (i.e., a total of 30 items). Each item was answered on a 5-point scale ranging from 0 (none at all) to 4 (a lot). The results were: tension (α = 0.785); anger (α = 0.849); depression (α = 0.772); fatigue (α = 0.841); vigour (α = 0.869); and friendliness (α = 0.804).

Resilience was measured using the Spanish validated version of the Brief Resilience Scale (BRS) [46], which consists of a six-question 5-point Likert scale ranging from 1 (totally disagree) to 5 (totally agree). The BRS reliability in our study was α = 0.736.

Motivation was measured using the Spanish validated version of the Sport Motivation Scale-II (SMS-II) questionnaire [38], which consists of 18 items covering six dimensions (three items per dimension), from 1 (does not correspond at all) to 7 (corresponds completely): intrinsic; integrated; identified; introjected; and external motivation; and amotivation. The results were: intrinsic (α = 0.715); integrated (α = 0.783); identified (α = 0.807); introjected (α = 0.622); and external motivation (α = 0.720); and amotivation (α = 0.470). Due to the low reliability values for introjected motivation and amotivation, both categories were excluded from the analysis.

### 2.3. Survey Distribution and Collection

The final version of the survey was written as a Google Forms questionnaire and was sent to players, coaches, and teams via social networks (Twitter, Facebook, and WhatsApp) using the snowball sampling technique [47]. In the social networks, a link to the questionnaire was published in the researcher’s profile. Additionally, direct messages were sent to those athletes, coaches and team leaders who met the inclusion criteria for the study. One week after sending out the questionnaire invitations, a follow-up was sent to increase the response rate. The questionnaire was available online from 16 April 2020 to 5 May 2020 under the state of alarm declared in Spain [2]. No surveys were accepted after the latter date. The dates were selected to allow comparisons with other team sports studies [4,12]. All participants signed an informed consent form before completing the survey. The questionnaire was open and anonymous to reduce potential bias (because players could not be identified through their responses), and unlimited time was provided for completion. After the deadline, the surveys were examined, and duplicate or contradictory responses rejected. In addition, it was checked that all the players met inclusion criteria of be a player federated in a national league during the survey.

### 2.4. Data Analysis

The data are described by arithmetic mean (*M*) and standard deviation (*SD*). The normal distribution of the variables was checked using the Kolmogorov–Smirnov and Shapiro–Wilk tests. To check the internal consistency and the reliability of the questionnaires Cronbach’s alpha test was used.

Paired sample t-tests were used to compare the periods before and during lockdown. Independent sample t-tests were performed to check gender differences. A two-way ANOVA was used to analyse the differences between level (professional and amateur), gender (male and female), and the interaction between them. To set the differences between groups, a post-hoc analysis was carried out using the Bonferroni test.

The effect size was estimated using Cohen’s d index (*d*) to analyse the periods before and during lockdown in groups with the same number of participants (men, women, professionals and non-professional) and Hedges’ *G* to analyse the isolation period in groups with different numbers of participants. Two cut-off points were established: medium effect (0.30) and large effect (0.60) [4,12]. The confidence interval for the effect size was set at 95%, and the percentage of change was calculated by (% change = ((M1 − M2)/M1) × 100).

Finally, a two-step hierarchical regression was performed to analyse the relationships between the psychological and training variables for the whole group. IBM SPSS Statistics software (SPSS 25.0. IBM Corp., Armonk, NY, USA) was used for the calculations. The level of significance was set at *p* < 0.05.

## 3. Results

### 3.1. Gender Differences

Differences by gender were found for fatigue (*p* = 0.022); friendship (*p* = 0.017); OEA (*p* < 0.001); and resilience (*p* = 0.031), with higher values for women with the exception of the latter. There were no gender differences in the other variables (*p* > 0.05; Table 1).

### 3.2. Comparisons of Time Period by Sport Level and Gender

For the whole group, RPE (*t*_168_ = 9.91; *p* < 0.001); training days (*t*_168_ = 2.95; *p* = 0.004); training hours (t_168_ = 13.66; *p* < 0.001); and sleep quality (*t*_168_ = 4.28; *p* < 0.001) were reduced during lockdown, while sleep hours (*t*_168_ = −5.52; *p* < 0.001) increased.

Similar results were obtained when the analysis was carried out according to gender. For the men, RPE (*t*_121_ = 9.45; *p* < 0.001); training days (*t*_121_ = 3.83; *p* < 0.001); training hours (*t*_121_ = 13.01; *p* < 0.001); and sleep quality (*t*_121_ = 2.88; *p* = 0.005) were reduced during lockdown, while sleep hours (*t*_121_ = −4.33; *p* < 0.001) increased. For the women RPE (*t*_46_ = 3.70; *p* = 0.001); training hours (*t*_46_ = 5.35; *p* < 0.001); and sleep quality (*t*_46_ = 3.38; *p* = 0.002) were reduced during the lockdown, while sleep hours (*t*_46_ = −3.44; *p* = 0.001) increased. However, training days did not show any change (*p* > 0.05).

Regarding the sport level, professional players reduced their RPE (*t*_53_ = 6.11; *p* < 0.001), training hours (*t*_53_ = 8.58; *p* < 0.001) and sleep quality (*t*_53_ = 3.05; *p* = 0.004) and increased the sleep hours (*t*_53_ = −3.73; *p* = 0.003) during lockdown. Contrary, training days did not show changes (*p* > 0.05). On the other hand, amateur players reduced their RPE (*t*_114_ = 7.90; *p* < 0.001), training days (*t*_114_ = 3.15; *p* = 0.002), training hours (*t*_114_ = 10.76; *p* < 0.001) and sleep quality (*t*_114_ = 3.05; *p* = 0.003) and increased the sleep hours (*t*_114_ = −4.57; *p* < 0.001) during lockdown (see Table 2).

### 3.3. Interactions between Gender and Sport Level in Each Period

Before lockdown, there were significant differences in training days (*F*_(3,165)_ = 3.74; *p* = 0.029) and training hours (*F*_(3,165)_ = 5.65; *p* = 0.001). Additionally, the post hoc analysis revealed the following differences between groups: For the whole group, professionals had higher RPE values (*p* = 0.035), train more days (*p* = 0.007), train more hours (*p* < 0.001), and had a better sleep quality (*p* = 0.041) than non-professionals. In non-professional, men had higher RPE values than women (*p* = 0.038). In men, professional train more hours than non-professionals (*p* = 0.003). In women, professionals had higher RPE values (*p* = 0.020), train more days (*p* = 0.026), train more hours (*p* = 0.010), and had a better sleep quality (*p* = 0.044) than non-professionals.

Regarding the lockdown period, there were significant differences in training days (*F*_(3,165)_ = 3.56; *p* = 0.016) and in training hours (*F*_(3,165)_ = 3.02; *p* = 0.031). The post hoc analysis revealed the following differences: professionals train more days than non-professionals (*p* = 0.012). In men, professionals train more days (*p* = 0.008) and more hours than non-professionals (*p* = 0.004). The rest of comparisons were not significant (*p* > 0.05) (See Table 3).

### 3.4. Results by Psychological Variables during Isolation Period

Rating of perceived exertion: the model was significant at step 1 (*F*_(2,166)_ = 4.30; *p* = 0.015; *r*^2^ = 0.049) and at step 2 (*F*_(12,156)_ = 3.17; *p* < 0.001; *r*^2^ = 0.196). Emotional intelligence was a significant predictor (*p* = 0.008, *β* = 0.24) at step 1 and depression (*p* = 0.001, *β* = −0.35) at step 2. The Δ*r*^2^ was significant from step 1 to step 2 (*p* = 0.003).

Training days: the model was significant at step 1 (*F*_(2,166)_ = 3.92; *p* = 0.022; *r*^2^ = 0.045) and at step 2 (*F*_(12,156)_ = 3.23; *p* < 0.001; *r*^2^ = 0.199). Emotional intelligence was a significant predictor (*p* = 0.016, *β* = 0.22) at step 1 and depression (*p* = 0.009, *β* = −0.27) at step 2 > 0. The Δ*r*^2^ was significant from step 1 to step 2 (*p* = 0.002).

Training hours: the model was significant at step 1 (*F*_(2,166)_ = 3.28; *p* = 0.040; *r*^2^ = 0.038) and at step 2 (*F*_(12,156)_ = 3.10; *p* = 0.001; *r*^2^ = 0.193). Emotional intelligence was a significant predictor (*p* = 0.035, β = 0.19) at step 1 and depression (*p* = 0.008, *β* = −0.28) and fatigue (*p* = 0.009, *β* = 0.27) at step 2. The Δ*r*^2^ was significant from step 1 to step 2 (*p* = 0.002).

Sleep variables: the model was neither significant at step 1 nor at step 2 (*p* > 0.05). In addition, according to sleep quality criterion, the model was significant at step 1 (*F*_(2,166)_ = 3.94; *p* = 0.014; *r*^2^ = 0.045) and at step 2 (*F*_(12,156)_ = 2.37; *p* = 0.008; *r*^2^ = 0.154). Emotional intelligence was a significant predictor (*p* = 0.014, *β* = 0.22) at step 1 and depression (*p* = 0.017, *β* = −0.26) at step 2. The Δ*r*^2^ was significant from step 1 to step 2 (*p* = 0.036) (see Table 4).

## 4. Discussion

The present study aimed to analyse the training habits and psychological states of Spanish basketball players of both genders and at different competitive levels during the COVID-19 lockdown. Overall, the results show that the confinement negatively affected their moods, and their training levels declined. However, players who showed high levels of EI trained more frequently and perceived themselves to be putting in more effort, in contrast with those who presented with high levels of depression. This suggests that, although confinement was an obstacle to maintaining training habits, athletes who were skilled at identifying and managing their emotions were more motivated to engage in physical activity.

### 4.1. Psychological Conditions

The emergency generated by the pandemic has led to an unprecedented worldwide crisis. Quarantine measures and their consequences have damaged many people‘s mental health [20]. The results of the present study show that, regardless of its duration, isolation had a significant impact on the mental states of the participants. Although lockdown was a novel phenomenon, the psychological problems it caused were similar to those experienced by athletes during longs periods of inactivity [48].

Overall, the participants showed moderate degrees of tension, depression, anger, and fatigue. Their negative moods were more intense than those of the athlete population in the original validation of the scale [40], which indicates that the participants were in a worse psychological state as a consequence of isolation. This is in keeping with previous studies [7]. However, the results also indicated a moderate level of resilience and adequate levels of EI, as well as high levels of intrinsic, integrated, and identified motivation. These findings suggest that, although the pandemic negatively impacted the participants’ moods, they were able to manage their emotions and retain a good level of motivation.

The women in the study presented with greater fatigue (understood as tiredness and low energy levels), but positive moods and dispositions towards others (i.e., friendship) and OEA than the men. On the other hand, men showed higher levels of resilience than the women. These findings align with the extensive literature indicating that women show greater cognitive or mood alterations than men [49]. This trend has also been maintained during the lockdown in the general population in different countries [23,24,25]. In Spain, these results are also evident both in the general population [50] and specifically in the sports context [51], where symptoms of mental fatigue have been more frequently associated with women.

It has also been widely argued that women express emotions more [52,53] and are more interpersonally orientated than men [54]. Women’s greater emotional distress for self and others may have an impact on resilience levels; lockdown may be traumatising them considerably more, which would pose an obstacle to psychological adaptation.

### 4.2. Training Conditions and Recovery

Regarding training habits before and after lockdown, the results indicate that all the players reduced their RPE, training days, and training hours. Some differences between gender and level were found. Women and professionals maintained their training days, while the men and the non-professionals did not. Only the male professionals trained more days and hours than the non-professionals during lockdown. Changes in training loads and sleep habits are in keeping with those found during the confinement period in other sports such as handball [4] or football [12]. Decreases in training levels are consistent with the consequences of quarantine, that is, the absence of competitions and training, a lack of communication between athletes and coaches, restricted access to equipment or space for exercise, and inappropriate training conditions [7]. A prolonged period of inactivity could be affecting the rate of degradation of contractile proteins, leading to muscle atrophy in the more extreme cases [55]. However, there is evidence that even after detraining or reduced training, the gains achieved in strength during regular basketball practice are maintained [56].

However, in terms of recovery, although players reported sleeping more hours, their sleep quality was worse during lockdown. Other studies have concluded that confinement has led to changes in athletes‘ sleep routines, including increased difficulties getting to sleep, daytime sleepiness, or sleeping later. These in turn increase the likelihood of higher rates of depression, anxiety, and stress [35]. Improved sleep, not only in terms of quantity but also quality and time, has been shown to have important implications for health and athletic performance [33,34], improvements in performance during matches [36], or in reaction times and mood [37].

Training and recovery during lockdown were predicted by EI and depression. Fatigue was only predictive of training hours. As sport is emotionally charged, knowing how to identify and manage emotions can lead to improved sporting performance [42]. Therefore, it is to be expected that during confinement, athletes with high levels of EI trained more days and hours and that they perceived themselves to be putting in more effort. Similar results were found by Mon-López [4,12] regarding handball and football players. These studies underlined that more emotionally intelligent people are more motivated to be physically active. Their training is less disrupted because they can control and manage their emotions more effectively. By contrast, depression, although reported to be lower in athletes than in non-athletes [31,32], was found to be a predictor of lower performance. This condition has been widespread during lockdown [20]. Isolation and a lack of social contact has led to the manifestation of symptoms such as anhedonia, a lack of energy, and increased fatigue [57].

### 4.3. Highlights and Limitations

Some practical implications can be derived from these results. In situations when training and competing are circumscribed, it is essential to adapt to the circumstances, resources, and particularities of the players. Plans should be designed not only to maintain physical activity but also to promote proper emotional management and understanding, by providing tools to control negative moods such as depression or mental fatigue. During lockdown, “quarantine camps” have proven to be a way to maintain sport-specific training. These training camps were designed to athletes maintain their sport-specific training with other athletes, with the support of coaches and support staff, minimising the risks of COVID-19 transmission. These have improved the athletes‘ mental and emotional health, training motivation, and perceived stress [58]. Jaenes Sánchez et al. [59] have found that coaching, support, and frequent training routines reduce some of the harmful effects of isolation on athletes‘ emotional well-being. Their findings, as well as the results of the present study, could be used by sports authorities to design policies and plans to help teams to offset the negative consequences of forced reductions in physical activity. It would be beneficial if these plans included measures to allow for continued team training supported by a multidisciplinary team including coaches, to guide sports training, physiotherapists, to prevent or treat any injuries, as well as sports psychologists, to teach emotional management and stress coping tools.

The present study has several limitations. The data collection was carried out online during lockdown, so it was not possible to control for the conditions under which the participants were completing the questionnaires. Completing them may have caused fatigue, and this may have influenced the results. Additionally, the final sample, although similar in size to other studies [4,12], comprised 169 players (after the injured, those infected with COVID-19, and those who gave contradictory answers were excluded). This may have limited the statistical power of the findings, so the data need to be treated with caution.

## 5. Conclusions

To the best of our knowledge, this is the first study conducted involving basketball players during the COVID-19 lockdown. The results suggest that the physical and psychological state of the athletes was affected during the lockout, especially in women. Although training volume was reduced during this period, professional athletes trained more days and hours than non-professional athletes. In addition, psychological variables such as adequate IE were found to be protective, limiting neither training conditions nor recovery. Further, negative moods such as depression seemed to predict lower sports performance.

## Figures and Tables

**Table 1 ijerph-18-09025-t001:** Basketball training variables and differences by gender during the isolation period.

Variables		All		Men		Women		*p*	Hedges’ *G*
		*M*	*SD*	*M*	*SD*	*M*	*SD*		
Descriptive	Days confined	34.53	3.70	34.64	3.79	34.23	3.48		
	Age	24.79	6.38	25.06	6.78	24.09	5.21		
	Living people	2.62	1.23	2.67	1.25	2.49	1.18		
	Living place	4.49	1.25	4.60	1.28	4.21	1.16		
Training	RPE	5.57	3.18	5.73	3.18	5.15	3.19		
	RPE-Isolation	3.80	2.71	3.76	2.73	3.91	2.69		
	Tdays	4.24	1.22	4.32	1.23	4.04	1.20		
	Tdays-Isolation	3.82	1.78	3.70	1.75	4.13	1.86		
	Thours	8.75	4.04	8.95	4.10	8.23	3.87		
	Thours-Isolation	5.00	3.25	4.92	3.37	5.21	2.93		
	Shours	7.24	0.92	7.24	0.88	7.26	1.01		
	Shours-Isolation	7.82	1.20	7.72	1.09	8.06	1.42		
	Squality	6.16	2.66	6.09	2.72	6.34	2.54		
	Squality-Isolation	5.36	2.69	5.48	2.75	5.02	2.53		
Moods	Tension	8.28	4.33	7.93	4.38	9.19	4.11		
	Depression	6.30	4.12	6.11	4.24	6.79	3.79		
	Anger	6.17	4.08	6.16	4.11	6.19	4.03		
	Vigor	12.89	3.98	12.74	4.00	13.28	3.96		
	Fatigue	6.95	4.18	6.49	4.22	8.13	3.89	0.022	0.396
	Friendship	15.18	3.15	14.82	3.28	16.11	2.60	0.017	0.414
Emotional intelligence	SEA	5.40	1.23	5.50	1.18	5.11	1.33		
	OEA	5.37	0.96	5.18	1.02	5.87	0.54	<0.001	0.766
	UOE	5.22	1.25	5.29	1.22	5.05	1.31		
	ROE	5.04	1.28	5.11	1.28	4.86	1.28		
	EI	21.03	3.29	21.08	3.30	20.89	3.31		
Resilience	BRS	3.41	0.73	3.48	0.71	3.22	0.74	0.031	−0.372
Motivation	Intrinsic	16.01	3.91	15.78	3.99	16.62	3.67		
	Integrated	16.85	3.78	16.66	3.85	17.34	3.57		
	Identified	17.37	3.40	17.22	3.54	17.74	3.03		
	External	8.07	4.61	8.24	4.61	7.62	4.65		

Notes: Days confined = number of days confined; Age (years); Living people = number of people with who player lives during the lockdown; Living place = place where player lives during the lockdown; RPE = rate of perceived exertion; RPE-Isolation = rate of perceived exertion during lockdown; Tdays = training days; Tdays-Isolation = training days during lockdown; Thours = training hours; Thours-Isolation = training hours during lockdown; Shours = sleep quantity (hours); Shours-Isolation = sleep quantity during lockdown (hours); Squality = sleep quality; Squality-Isolation = sleep quality during lockdown; SEA = self-emotion appraisal; OEA = other’s emotion appraisal; UOE = use of emotion; ROE = regulation of emotion; BRS = brief resilience scale.

**Table 2 ijerph-18-09025-t002:** Differences on training variables between pre-isolation and isolation period by gender and sport level.

		Pre-Isolation		Isolation Period		*r*	*p*	Cohen’s *d*			Interval Confidence 95%		% Change
		*M*	*SD*	*M*	*SD*			*d*	*d* Pooled	*d* Ind Group	*LL*	*UL*	
ALL	RPE	5.57	3.18	3.80	2.71	0.70	<0.001	−0.719	−0.773	−0.557	−0.939	−0.499	−46.35
	Tdays	4.24	1.22	3.82	1.78	0.29	0.004	−0.289	−0.231	−0.344	−0.503	−0.075	−10.99
	Thours	8.75	4.04	5.00	3.25	0.54	<0.001	−0.968	−1.066	−0.928	−1.19	−0.742	−75.03
	Shours	7.24	0.92	7.82	1.20	0.20	<0.001	0.498	0.429	0.63	0.282	0.715	7.34
	Squality	6.16	2.66	5.36	2.69	0.58	<0.001	−0.328	−0.326	−0.301	−0.543	−0.113	−15.03
MEN	RPE	5.73	3.18	3.76	2.73	0.71	<0.001	−0.813	−0.873	−0.619	−1.04	−0.592	−52.29
	Tdays	4.32	1.23	3.70	1.75	0.33	<0.001	−0.435	−0.354	−0.504	−0.651	−0.22	−16.59
	Thours	8.95	4.10	4.92	3.37	0.60	<0.001	−1.10	−1.20	−0.983	−1.33	−0.87	−82.00
	Shours	7.24	0.88	7.72	1.09	0.23	<0.001	0.44	0.39	0.545	0.224	0.655	6.26
	Squality	6.09	2.72	5.48	2.75	0.64	0.005	−0.264	−0.263	−0.224	−0.478	−0.05	−11.06
WOMEN	RPE	5.15	3.19	3.91	2.69	0.71	0.001	−0.458	−0.495	−0.389	−0.868	−0.049	−31.52
	Tdays	4.04	1.20	4.13	1.86	0.21	0.770						2.06
	Thours	8.23	3.87	5.21	2.93	0.38	<0.001	−0.701	−0.79	−0.78	−1.12	−0.284	−57.96
	Shours	7.26	1.01	8.06	1.42	0.16	0.001	0.611	0.501	0.792	0.197	1.03	10.03
	Squality	6.34	2.54	5.02	2.53	0.44	0.002	−0.491	−0.492	−0.52	−0.901	−0.081	−26.27
Professionals	RPE	6.07	3.06	4.39	2.75	0.76	<0.001	−0.796	−0.837	−0.549	−1.188	−0.404	−38.40
	Tdays	4.56	1.06	4.43	1.74	0.30	0.588						−2.93
	Thours	10.39	4.27	5.89	3.72	0.54	<0.001	−1.1	−1.173	−1.054	−1.505	−0.695	−76.41
	Shours	7.43	0.94	7.96	1.13	0.25	0.003	0.459	0.415	0.564	0.077	0.841	6.74
	Squality	6.69	2.32	5.57	2.41	0.36	0.004	−0.427	−0.419	−0.483	−0.808	−0.045	−19.93
No- Professionals	RPE	5.33	3.22	3.53	2.66	0.67	<0.001	−0.689	−0.751	−0.559	−0.955	−0.423	−50.99
	Tdays	4.10	1.27	3.54	1.74	0.24	0.002	−0.357	−0.298	−0.441	−0.618	−0.097	−15.73
	Thours	7.98	3.71	4.58	2.93	0.50	<0.001	−0.915	−1.015	−0.916	−1.186	−0.643	−74.19
	Shours	7.16	0.89	7.75	1.23	0.17	<0.001	0.515	0.427	0.663	0.252	0.778	7.63
	Squality	5.91	2.79	5.25	2.82	0.66	0.003	−0.285	−0.284	−0.237	−0.545	−0.025	−12.58

Notes: RPE = rate of perceived exertion; Tdays = training days; Thours = training hours; Shours = sleep quantity (hours); Squality = sleep quality; r = Pearson level of correlation between isolation periods; *p* = level of significance; *d* = effect size. LL = lower limit; UL = upper limit.

**Table 3 ijerph-18-09025-t003:** Differences by gender and sport level in each period.

		Prof	N-Prof	*p*	Men	Women	*p*	Prof		*p*	No Prof		*p*	Men		*p*	Women		*p*
								Men	Women		Men	Women		Prof	No Prof		Prof	No Prof	
Pre isolation period	RPE	6.14	4.95	**0.035**	5.78	5.31	0.409	5.88	6.40	0.560	5.67	4.22	**0.038**	5.88	5.67	0.739	6.40	4.22	**0.020**
	Tdays	4.54	3.96	**0.007**	4.40	4.10	0.164	4.59	4.50	0.795	4.22	3.70	0.054	4.59	4.22	0.126	4.50	3.70	**0.026**
	Thours	10.30	7.63	**<0.001**	9.47	8.46	0.146	10.65	9.95	0.525	8.30	6.96	0.121	10.65	8.30	**0.003**	9.95	6.96	**0.010**
	Shours	7.43	7.14	0.078	7.29	7.28	0.949	7.41	7.45	0.882	7.17	7.11	0.769	7.41	7.17	0.194	7.45	7.11	0.211
	Squality	6.80	5.83	**0.041**	6.17	6.46	0.545	6.35	7.25	0.231	5.99	5.67	0.581	6.35	5.99	0.497	7.25	5.67	**0.044**
Isolation period	RPE	4.37	3.57	0.097	3.97	3.97	0.990	4.44	4.30	0.853	3.50	3.63	0.828	4.44	3.50	0.087	4.30	3.63	0.402
	Tdays	4.44	3.65	**0.012**	3.91	4.18	0.400	4.38	4.50	0.811	3.44	3.85	0.289	4.38	3.44	**0.008**	4.50	3.85	0.210
	Thours	5.75	4.80	0.100	5.34	5.21	0.821	6.29	5.20	0.225	4.39	5.22	0.235	6.29	4.39	**0.004**	5.20	5.22	0.981
	Shours	8.00	7.84	0.438	7.76	8.08	0.145	7.85	8.15	0.380	7.67	8.00	0.212	7.85	7.67	0.451	8.15	8.00	0.671
	Squality	5.64	4.95	0.148	5.45	5.14	0.508	5.38	5.90	0.493	5.52	4.37	0.052	5.38	5.52	0.795	5.90	4.37	0.054

Notes: Prof = professional players; N-Prof = non-professional players; RPE = rate of perceived exertion; Tdays = training days; Thours = training hours; Shours = sleep quantity (hours); Squality = sleep quality; *p* = level of significance. Significant correlations are in bold letters.

**Table 4 ijerph-18-09025-t004:** Hierarchical regressions of training variables onto the psychological factors (emotional intelligence, resilience, moods, and motivations) of basketball players during lockdown.

Model	Predictor	RPE			T-Days			T-Hours			S-Hours			S-Quality		
		*β*	*t*	*p*	*β*	*t*	*p*	*β*	*t*	*p*	*β*	*t*	*p*	*β*	*t*	*p*
Step 1	EI	0.24	2.67	0.008	0.22	2.43	0.016	0.19	2.12	0.035	0.13	1.42	0.157	0.22	2.49	0.014
	BRS	−0.03	−0.35	0.728	−0.01	−0.08	0.934	0.01	0.12	0.907	−0.18	−1.94	0.054	−0.02	−0.20	0.844
	F/R^2^/Adj. *R*^2^	4.30/0.049/0.038		0.015	3.92/0.045/0.034		0.022	3.28/0.038/0.026		0.04	2.00/0.024/0.012		0.138	3.94/0.045/0.034		0.021
Step 2	EI	0.08	0.88	0.380	0.09	0.99	0.324	0.10	1.08	0.284	0.19	1.83	0.069	0.15	1.57	0.119
	BRS	−0.11	−1.13	0.260	−0.05	−0.53	0.598	0.01	0.09	0.931	−0.21	−2.07	0.040	−0.10	−1.07	0.285
	Tension	−0.03	−0.28	0.781	0.11	1.01	0.313	−0.07	−0.63	0.531	0.07	0.58	0.564	0.07	0.66	0.514
	Depression	−0.35	−3.41	0.001	−0.27	−2.64	0.009	−0.28	−2.70	0.008	−0.26	−2.35	0.020	−0.26	−2.42	0.017
	Anger	0.09	0.79	0.431	0.14	1.21	0.228	0.23	1.97	0.051	0.02	0.17	0.862	−0.03	−0.24	0.813
	Vigor	0.14	1.24	0.216	0.08	0.69	0.489	0.18	1.57	0.119	−0.03	−0.26	0.798	0.03	0.26	0.796
	Fatigue	0.08	0.75	0.458	−0.01	−0.10	0.92	0.27	2.64	0.009	0.10	0.94	0.349	−0.15	−1.46	0.147
	Friendship	−0.05	−0.59	0.559	−0.04	−0.50	0.622	0.03	0.31	0.761	−0.16	−1.71	0.089	−0.09	−1.01	0.314
	Intrinsic	0.13	1.05	0.293	0.17	1.42	0.158	0.13	1.06	0.29	0.05	0.40	0.691	−0.03	−0.21	0.838
	Integrated	0.07	0.53	0.599	0.25	2.01	0.046	0.19	1.47	0.143	0.02	0.17	0.863	−0.17	−1.29	0.2
	Identified	0.05	0.36	0.723	−0.09	−0.73	0.470	−0.08	−0.64	0.526	−0.10	−0.68	0.498	0.12	0.86	0.391
	External	−0.04	−0.45	0.654	−0.13	−1.67	0.097	−0.05	−0.57	0.57	0.05	0.59	0.559	0.00	0.00	0.998
	F/R^2^/Adj. *R*^2^	3.17/0.196/0.134		<0.001	3.23/0.199/0.138		<0.001	3.10/0.193/0.131		0.001	1.14/0.081/0.010		0.326	2.37/0.154/0.089		0.008
	Δ F/Δ *R^2^*	2.84/0.147		0.003	3/0.154		0.002	2.99/0.155		0.002	0.98/0.058		0.47	2.01/0.109		0.036

Notes: EI = Emotional intelligence; BRS = brief resilience scale; RPE = rate of perceived exertion; Tdays = training days; Thours = training hours; Shours = sleep quantity; Squality = sleep quality; Moods = tension, depression, anger, vigour, fatigue and friendship; Motivations = intrinsic, integrated, identified, and external; R^2^ = R-squared value; *p* = level of significance.

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
