# Peer review of "Psychological States and Training Habits during the COVID-19 Pandemic Lockdown in Spanish Basketball Athletes"

_ijerph, 2021, doi:10.3390/ijerph18179025_

Round 1

Reviewer 1 Report

Please be more explicit in the conclusion section in accordance with the aim of the paper.

Author Response

Dear revisor,

Thank you for your comment. We have been more explicit in the conclusions as you suggested, responding to the objective of the study. 

We hope we have answered your questions.

Thank you very much.

Reviewer 2 Report

112-114.

Should specify the competitions considered as professional and non-professional leagues.

124-125.

The inclusion of the questionnaire raised in the investigation would also be of interest.

115-118.

Data reference for gender of participants is missing.

125-127.

It would be interesting if they included the criteria for adapting the relevant questionnaires.

144-145.

The statistic used to obtain these results should be indicated.

166-167.

It would be interesting to know when the "launch" of the questionnaire was proposed and the stages leading up to it. Indicating how contacts with athletes were made and ways for the "snowball" to spread.

171-173.

The questionnaire was open and anonymus to reduce potential biases (because player could not be identified through their answers). This is precisely why it is important to explain in detail the procedure and the criteria for selecting participants.

185.

I consider it appropriate to specify which groups have the "same number" of participants, for the purposes indicated in the calculation of the effect size.

297-303

The articles used to discuss the results should be expanded to include groups of characteristics similar to those of the proposed work. The proposed references are based on population aged [24] between 18-74 years and with 85% women and [26] between 18-64 years and with 55% women, in addition to the possible differences due to the cultural context and the patterns of coexistence established in confinement.

Author Response

Dear Reviewer,

Thank you very much for your comments which have helped to improve the quality of the manuscript. After reviewing and incorporating the changes, we respond below to the suggestions provided.

  • 112-114. Should specify the competitions considered as professional and non-professional leagues.
    Response: Thanks for the suggestion. The competitions have been added. Professional players were considered to be those players who participated in the first and second national leagues (ACB and LEB Gold for men and Liga Femenina 1 and 2 for women), while amateur players were those who played in the minor national leagues (LEB Silver, EBA and 1ª Nacional for men and 1ª Nacional for women).

  • 124-125. The inclusion of the questionnaire raised in the investigation would also be of interest.
    Response: Thank you. The questionnaire has been added. Please see the attachment.

  • 115-118. Data reference for the gender of participants is missing.
    Response: the next reference has been added “CSD. Estadistica de Deporte Federado 2019. Técnica, S.G., Ed. Ministerio de Cultura y Deporte: Madrid, 2020.”

  • 125-127. It would be interesting if they included the criteria for adapting the relevant questionnaires.
    Response: Thank you. The sentence has been modified to clarify what items were adapted to basketball: “Demographic questions which were specific for basketball sport (sport level and play position) were adapted from Mon-López et al. [12, 13] by two PhD professional basketball coaches.”

  • 144-145. The statistic used to obtain these results should be indicated.
    Response: Thank you for this commentary. We have added a sentence in the data analysis subsection explaining what we have done: “To check the internal consistency and the reliability of the questionnaires Cronbach’s alpha test was used.” Demographic questions which were specific for basketball sport (sport level and play position) were adapted from Mon-López et al. [12, 13] by two PhD professional basketball coaches.

  • 166-167. It would be interesting to know when the "launch" of the questionnaire was proposed and the stages leading up to it. Indicating how contacts with athletes were made and ways for the "snowball" to spread.
    Response: Thank you for your suggestion. The following paragraph has been added: “In the social networks, a link to the questionnaire was published in the researcher’s profile. Additionally, direct messages were sent to those athletes, coaches, and team leaders who met the inclusion criteria for the study.” To check the internal consistency and the reliability of the questionnaires Cronbach’s alpha test was used.

  • 171-173. The questionnaire was open and anonymus to reduce potential biases (because players could not be identified through their answers). This is precisely why it is important to explain in detail the procedure and the criteria for selecting participants.
    Response: Thank you for your suggestion. We have added a sentence explaining the selection of the participants: “In addition, it was checked that all the players met inclusion criteria of being a player federated in a national league during the survey.”

  • 185. I consider it appropriate to specify which groups have the "same number" of participants, for the purposes indicated in the calculation of the effect size.
    Response: Thank you for your suggestion. According to what we wrote in line 179 “Paired sample t-tests were used to compare the periods before and during the lockdown.” we have modified the text as follows: “The effect size was estimated using Cohen’s d index (d) to analyse the periods before and during the lockdown in groups with the same number of participants (men, women, professionals and non-professional)”

  • 297-303. The articles used to discuss the results should be expanded to include groups of characteristics similar to those of the proposed work. The proposed references are based on population aged [24] between 18-74 years and with 85% women and [26] between 18-64 years and with 55% women, in addition to the possible differences due to the cultural context and the patterns of coexistence established in confinement.
    Response: Thank you very much for your suggestion. Given the novelty of the subject matter, it is very difficult to find similar studies in the same population, with samples of equivalent characteristics and in the same cultural context. However, we have included new evidence that indicates that in Spain, in the general population, women were also more affected psychologically, as well as in the sports context.

Since you indicate that "I don't feel qualified to judge about the English language and style", please see attached the certificate of Proof-Reading-Service which ensures the language proofreading.

Reviewer 3 Report

I would like to thank the editorial board for the opportunity to review the current manuscript that was titled “Psychological states and training habits during the COVID-19 2

pandemic lockdown in Spanish basketball athletes”. The current paper makes a valuable contribution to understand the psychological and training impacts that lockdown had on both professional and semi-professional players.

General Comment

The manuscript is well presented with informative tables to present the data. Even though this study is novel and provides an interesting overview of what occurred during the lockdown, I feel that the practical applications are weak. The authors, with their new knowledge need to provide further information so that coaches and players can learn from the experiences encountered during this lockdown.

I have some further minor comments which are outlined below:

Introduction

Overall the introduction is clear for the reader to understand the background for the study. Some minor edits are required:

L51-53 Additional information is required here. GPS records the players movements, it can record heart rate but requires a heart rate monitor to be linked to the GPS unit. As written, it is not clear.

L57 – you mention a “higher risk”.... please indicate what you mean here, a higher risk of??

L65 – you need to include a reference for this statement

Methods

L107 & 119-120 – You mention that you removed 1 person who was infected with Covid, the in 119-120 you mention that 5 players recovered from covid during the study? What was the difference between these players? Did this 1 person have covid as the questionnaires were handed out and the 5 developed covid after the questionnaires were handed out??  Can you clarify in the text

L188-189 – you have included the calculation within the brackets? Please update

Discussion

L 349 – you mentioned “quarantine camps”....please provide details of these camps as you mentioned that they are proven to be a way of maintaining sport-specific training. It is important to explain as this could be interpretated as training together which was forbidden during the restrictions

Please provide more details for your practical applications. You mentioned that “plans should be designed not only to maintain physical activity but also to promote proper emotional management and understanding, by providing tools to control negative moods such as depression or mental fatigue” For coaches to learn from your study, please provide what they can do, provide some examples.

Author Response

Dear Reviewer,

Thank you very much for your comments which have helped to improve the quality of the manuscript. After reviewing and incorporating the changes, we respond below to the suggestions provided.

  • L51-53 Additional information is required here. GPS records the players movements, it can record heart rate but requires a heart rate monitor to be linked to the GPS unit. As written, it is not clear.
    Response: The term ‘heart rate’ has been replaced by the generic concept of ‘external load’ due to the fact that it collects more broadly all the physical load data that can be controlled through GPS.

  • L57 – you mention a “higher risk”.... please indicate what you mean here, a higher risk of??
    Response: The indicated sentence has been removed from the paragraph because the risk of using individualised training methodologies is set out above in the first sentence.

  • L65 – you need to include a reference for this statement
    Response: The reference has been added (% change = [(M1 - M2) / M1] * 100).

  • L107 & 119-120 – You mention that you removed 1 person who was infected with Covid, the in 119-120 you mention that 5 players recovered from covid during the study? What was the difference between these players? Did this 1 person have covid as the questionnaires were handed out and the 5 developed covid after the questionnaires were handed out??  Can you clarify in the text
    Response: Thank you for your appreciation. We have tried to clarify this issue modifying the following sentence: “Finally, five players had previously been infected with COVID-19 and when they filled out the questionnaire they were already fully recovered from the disease.”

  • L188-189 – you have included the calculation within the brackets? Please update
    Response: The formula has been amended: “(% change = [(M1 - M2) / M1] * 100).” Finally, five players had previously been infected with COVID-19 and when they filled out the questionnaire they were already fully recovered from the disease.

  • L 349 – you mentioned “quarantine camps”....please provide details of these camps as you mentioned that they are proven to be a way of maintaining sport-specific training. It is important to explain as this could be interpretated as training together which was forbidden during the restrictions
    Response: Thank you for your comment. We have clarified that the quarantine camps were training camps specifically designed during the pandemic so that athletes could continue to train with other athletes while minimising the risk of COVID-19 infection.

    Please provide more details for your practical applications. You mentioned that “plans should be designed not only to maintain physical activity but also to promote proper emotional management and understanding, by providing tools to control negative moods such as depression or mental fatigue” For coaches to learn from your study, please provide what they can do, provide some examples.
    Response: We have indicated that to reduce the negative psychological impact, these plans have to be designed by a multidisciplinary team, including sports psychologists. They should be the ones to offer tools to reduce negative emotions and possible stress or depression.

Since you indicated that "English language and style are fine/minor spell check required", please see attached the certificate from the proofreading service which ensures that the document has been proofread to ensure language consistency.
